# Current and Future Methodology for Quantitation and Site-Specific Mapping the Location of DNA Adducts

**DOI:** 10.3390/toxics10020045

**Published:** 2022-01-19

**Authors:** Gunnar Boysen, Intawat Nookaew

**Affiliations:** 1Department Environmental and Occupational Health, Fay W. Boozman College of Public Health, University of Arkansas for Medical Sciences, Little Rock, AR 72205, USA; 2The Winthrop P. Rockefeller Cancer Institute, University of Arkansas for Medical Sciences, Little Rock, AR 72205, USA; INookaew@uams.edu; 3Department Biomedical Informatics, College of Medicine, University of Arkansas for Medical Sciences, Little Rock, AR 72205, USA

**Keywords:** DNA adducts, nanopore, Oxford Nanopore Technology, mass spectrometry, adductomics, exposome

## Abstract

Formation of DNA adducts is a key event for a genotoxic mode of action, and their presence is often used as a surrogate for mutation and increased cancer risk. Interest in DNA adducts are twofold: first, to demonstrate exposure, and second, to link DNA adduct location to subsequent mutations or altered gene regulation. Methods have been established to quantitate DNA adducts with high chemical specificity and to visualize the location of DNA adducts, and elegant bio-analytical methods have been devised utilizing enzymes, various chemistries, and molecular biology methods. Traditionally, these highly specific methods cannot be combined, and the results are incomparable. Initially developed for single-molecule DNA sequencing, nanopore-type technologies are expected to enable simultaneous quantitation and location of DNA adducts across the genome. Herein, we briefly summarize the current methodologies for state-of-the-art quantitation of DNA adduct levels and mapping of DNA adducts and describe novel single-molecule DNA sequencing technologies to achieve both measures. Emerging technologies are expected to soon provide a comprehensive picture of the exposome and identify gene regions susceptible to DNA adduct formation.

## 1. Introduction

Although DNA is a very stable molecule for storing biological information, it is under relentless attack by reactive compounds of endogenous and exogenous origin that covalently bind to DNA, forming so called DNA adducts [1]. Since identification of the first DNA adduct by Reiner and Zamenhof in 1957 [2], several thousand studies on DNA adducts have been reported and were reviewed from different perspectives [3,4,5,6,7,8,9,10]. The ability of a compound to form DNA adducts, directly or after metabolic activation, is considered a critical event in chemical carcinogenesis [11] and a key event for the genotoxic mode of action of toxicants [12,13]. The binding to DNA has been widely used as biomarker of exposure in molecular epidemiology studies to link exposure to adverse health outcomes [14,15,16]. Formation and stability of specific promutagenic DNA adducts has been established in vitro, in cell cultures, animal experiments, and molecular epidemiology studies [5,6,17]. Recent advances in technology, especially in mass spectrometry, allows for ‘omics’ type monitoring of DNA adducts, DNA adductomics, and is expected to provide unprecedented insight into the total exposome [18,19,20,21]. 

This communication summarizes the current methodologies for quantitation of DNA adduct levels and mapping the location of DNA adducts along the genome and highlights novel single-molecule analysis expected to be capable of achieving both measures. 

## 2. Methodology 

Historically, studies on DNA adducts have mirrored advances in analytical and bioanalytical chemistry technologies, starting with paper chromatography, enzyme-linked immunosorbent assay [22,23] and ^32^P-Post-labeling [24] and followed by liquid [25] and gas chromatography separations with various detection systems, including UV, fluorescence, and electrochemical detectors [26,27].

### 2.1. Mass Spectrometry-Based DNA Adduct Quantitation

The introduction of electrospray ionization by Dole and others in 1968 [28] and its subsequent application to proteins by Fenn et al. in 1990 revolutionized biochemistry [29]. Electrospray ionization coupled with mass-spectrometry is now a commonly used technique for qualitative and quantitative analyses of many types of compounds, including DNA adducts [26,27]. Improvements in mass analyzers allows monitoring of thousands of molecules simultaneously with ultra-high mass resolution and accuracy [30,31,32,33]. The basic and most frequently used approach for quantitation of DNA adducts is to isolate DNA, liberate the DNA adducts from DNA by chemical means or enzyme hydrolysis, and quantitate the released 2′-deoxyribonucleoside- and nucleobase-adducts by LC-MS (Figure 1a) [34]. To improve sensitivity and measurement accuracy, various sample enrichment procedures maybe included such as solid phase and liquid–liquid extraction or pre-separation by HPLC [3], and stable isotope standards are added at the beginning to account for any potential loss during sample workup [35,36,37,38].

### 2.2. Adductomics, Nontargeted and Qualitative Screening of DNA Adducts

While most studies apply targeted mass spectrometry using authentic DNA adduct standards, efforts are underway to move towards nontargeted ‘omics’ type screening of DNA adducts to obtain a complete measure of the exposome [39]. Therefore, DNA adducts are monitored in various modes of data-dependent or multistage (MS^n^) scanning modes [19,40]. DNA adductomics methods have been reviewed extensively [18,20,32,37].

Independent of a targeted or nontargeted approach, results are reported as number or concentration of DNA adducts per DNA, or the corresponding unmodified 2′-deoxyribo-nucleosides (e.g., fmol/mg DNA, fmol/µmol dG). The mass spectrometry-based detection provides a high level of chemical specificity but no site-specificity.

### 2.3. Amplification-Based Mapping of DNA Adducts

Various methods have been developed for genome-wide and site-specific mapping of DNA damage. The general approach is based on mapping sequencing stop sites to localize the adducts [41,42,43]. Therefore, DNA adducts are first recognized or modified by enzymatic or chemical means, taking advantage of DNA repair enzymes to mark and excise DNA adducts (Figure 1b). The marked or cleaved DNA is then amplified and sequenced and the location of DNA adducts are obtained from strand ends or mismatched base pairs [41,42,43]. 

For example, Denissenko et al. used UvrABC excision nuclease, in combination with ligation-mediated polymerase chain reaction (PCR) to map the sites of bulky DNA adducts [44,45,46]. More recently, cyclobutene pyrimidine dimers (CPDs) were mapped using immunoprecipitation based DIIP assay [21] or CPD-seq utilizing DNA cleavage mediated by T4 endonuclease V and APE1 [47,48]. XR-seq [49] and tXR-seq rely on TFIIH-mediated enrichment of damage-containing fragments cleaved by mammalian nucleotide excision repair enzymes [50,51,52]. Cisplatin-seq takes advantage of the HMG box A of HMGB1 protein’s preferentially binding to distorted DNA structures for selective enrichment of cisplatin-modified DNA [53]. Further, click chemistry, Click-Code-Seq, has been successfully applied to label 8-oxo-7,8-dihydroguanine (8-oxo-dG) or 5-hydroxymethylcytosine (5hmdC) sites in DNA prior to next-generation sequencing [54,55]. DNA adduct mapping methods have been reviewed extensively [56,57,58]. 

Results are given as modified versus unmodified sites or motifs with high level of site-specificity. Unfortunately, these elegant methods are limited by (i) the breadth of enzyme specificity, which may identify a mixture of DNA adducts; (ii) excision of short DNA fragments that sometimes cannot be aligned with absolute certainty; (iii) reliance on completion of chemical reactions; (iv) restricted applicability to one type or class of DNA adducts; (v) the inability to distinguish different DNA adducts; (vi) limited chemical specificity; and (vii) lack of antibodies with suitable specificity for the adducts of interest. 

### 2.4. Single Molecule DNA Sequencing 

A relatively new method for analysis of DNA adducts is based on single molecule sequencing. This novel technology sequences and counts single DNA molecules, whether they are whole genomes or DNA fragments. 

#### 2.4.1. Nanopore Technology

Nanopore-type technology utilizes electrochemical forces to pull single-stranded DNA in native form through tiny pores (Figure 1c). The accompanying changes in electric current indicate the physicochemical properties of the DNA nucleobases transiting through the pore, revealing the DNA sequence and potential DNA adducts [59]. A DNA adduct modulates the nanopore ion current signal while entering, passing through, and exiting the nanopore. This results in an electric current signature characteristic for the DNA adduct within a given 7-base sequence that includes the DNA adduct and the three adjacent 3′ and 5′ nucleotides that reside in the nanopore [59].

Burrows and colleagues pioneered the application of nanopore-type technology for sequencing DNA adducts in single-stranded DNA. With custom-made solid-state or protein-based nanopores, they showed the proof-of-principle for detecting *N*^2^-BPDE-dG-induced adducts [60], abasic sites [61,62,63], and other DNA adducts [64], including 8-oxo-dG [65,66,67].

##### Oxford Nanopore Technologies (ONT) 

Using a similar principle, ONT developed and commercialized a nucleic acid sequencing technology that has the capability to sequence long to ultra-long molecules of DNA (>2 Mb) in the native form, preserving the sequence position of the DNA adducts [68,69,70]. 

##### PacBio DNA Sequencer

While technically not a nanopore system, the PacBio RSII DNA sequencer also employs the single-molecule real-time (SMRT) sequencing principle. The DNA strands are converted into loops and amplified by a polymerase. The unique time needed by the polymerase for elongation of the DNA is indicative of the base added and DNA sequence. 

#### 2.4.2. Data Types and Analyses 

While ONT uses the disturbance in the ion signals, the PacBio system makes use of the time delay of the polymerase to identify the DNA sequence. The standard base calling algorithms for both systems are optimized for the four main nucleobases (A, T, G, and C), and report errors when encountering unknown bases, potential DNA adducts. Therefore, efforts are under way to expand base calling algorithms to enable detection of DNA adducts and epigenetic modifications. Multiple groups successfully applied ONT for genome-wide detection of epigenetic modifications, such as 5-methyl-2′-deoxycytidine (5 mdC) and *N*^6^-methyl-2′-deoxyadenosine (*N*^6^ mdA) [71,72,73,74]. Similarly, the PacBio system has been shown to be suitable for simultaneous detection of *N*^6^ mdA, 5mdC, and 5hmdC [75]. The latter, is also capable for detection of the unique phosphothioate modifications of the phospho-ribose backbone, found in some bacteria [76,77]. Expansion of these commercial platforms to high abundant epigenetic marks is the first step to enable them to detect any DNA modifications, including DNA adducts derived from endogenous or exogenous sources. 

##### ONT/ELIGOS

Our team developed the Epitranscriptional/Epigenomical Landscape Inferring from Glitches of ONT Signals (ELIGOS) software that uses ONT data to simultaneously detect RNA and DNA modifications, including DNA adducts [59,78,79]. The ONT/ELIGOS platform is a powerful tool for (i) detecting DNA adducts and (ii) discriminating DNA adducts of different sizes, regiochemistry, and functional groups [59]. ELIGOS takes the error information from the standard base-calling algorithm and calculates the odds ratios at each site as an indicator for a potential DNA adduct (Figure 1) [59].

Next, ELIGOS generates a radar plot displaying the multiplex disturbances of the re-squiggled ONT signal from the DNA adduct in relationship to the dG-containing control plasmid (Figure 2a) [59]. The radar plot shows an 11-base sequence that covers the DNA adduct along with five preceding and five trailing deoxynucleotides that dwell in the nanopore during sequencing [59,80]. These radar plots are characteristic of each DNA adduct at each given position. The radar plots are used to identify the type of DNA adduct by comparing them with plots of standard DNA that contains the DNA adduct at the same sequence position and within the same sequence context (Figure 2a). Disturbances in the raw ONT signal are used for multivariant statistical analyses to obtain a measure of separation, as shown in Figure 2b.

#### 2.4.3. CRISPR/cas-9 Targeted Sequencing

DNA adduct measurement by single molecule sequencing technologies will generate a huge amount of data because the DNA adduct levels are extremely low. In principle, one would need to sequence the whole genome 10^8^ times to get a DNA adduct level at each site of 1 adduct/10^8^ unmodified 2′-deoxyribonucleosides (nnt) as commonly reported. However, drawing from previous mapping approaches using CRISPR/cas-9, targeted sequencing will increase the number of reads and thereby sensitivity at the sites of interest such as mutation hotspots or cancer driver genes [81,82,83]. The CRISPR/cas-9 targeted sequencing may target DNA segments of 1000 to 10,000 base pairs in length that can be read by ONT as a single molecule.

#### 2.4.4. Limit of Detection

The results of these single molecule analyses are number of DNA adduct X at Site Y per total number of DNA molecules / DNA molecules analyzed containing the site of interest, including unmodified DNA and DNA molecules carrying mutations or modification at different sites. In theory, a DNA segment of 2000 base pairs length that has been read 100,000 times has a limit of detection of

one adduct at a given site/per 10^5^ unmodified bases at position Y (e.g., alkyl-dG/10^5^ dG at position Y);five adducts per 10^9^ unmodified nucleotides (e.g., alkyl-dG/10^9^ nnt);or approximately one adduct per 10^9^ corresponding nucleotides (e.g., alkyl-dG/10^9^ dG).

These theoretical limits of detection of ONT/ELIGIOS are in the range of the levels reported for endogenous and exogenous DNA adducts.

## 3. Discussion 

Below we highlight some selected studies showing the application and utility of DNA adduct research. 

### 3.1. DNA Adduct Levels

The first question in the DNA adduct field was, and still is, to demonstrate the formation of exposure-induced DNA adducts. For example, benzo[*a*]pyrene (B[*a*]P), a ubiquitous environmental and occupational carcinogen and a tobacco smoke constituent has been studied extensively. B[*a*]P requires metabolic activation to the reactive 7,8,-diepoxy-9,10-dihydroxy-benzo[*a*]pyrene (BPDE) that ultimately forms the promutagenic *N*^2^-BPDE-dG adduct. Applying an ultra-sensitive LC-MS method, with a limit of detection of 1 *N*^2^-BPDE-dG adduct per 10^11^ nucleotides (1 adduct per about 30 human cells) levels in lung DNA of smokers and nonsmokers were 3.1 and 1.3 *N*^2^-BPDE-dG adducts per 10^11^ nucleotides, respectively [84]. 

With increases in sensitivity of the methodologies, it became apparent that DNA adducts are also formed by reactive compounds of endogenous origin, such as the *N*7-(2-hydroxyethyl) guanine [85,86], 1,*N*^6^-ethenodeoxyadenosine(edA), 3,*N*^4^-ethenodeoxycytosine (εdC), and *N*^2^,3-ethenoguanine(εG) [87,88]. In addition, oxidatively generated modifications of DNA have been shown to be present in all cellular DNA [89,90]. The discrepancies in levels of 8-oxo-dG determined by different methods were attributed to artifacts leading to an over-estimation of the true levels. To address this issue, the European Standards Committee on Oxidative DNA Damage (ESCODD) initiated an elegant multi-laboratory comparison study to build consensus [91,92]. Ten laboratories quantitated 8-oxodG in DNA from HeLa cells and established a background level of 1 to 4 8-oxo-dG per 10^6^ dG [92,93]. Endogenous 8-oxo-dG in human peripheral blood lymphocytes measured by LC-MS are about one 8-oxo-dG per 10^6^ dG [89,94,95]. Consequently, endogenous 8-oxo-dG levels are 5 orders of magnitude higher than the exposure-derived *N*^2^-BPDE-dG [84]. 

Most studies are on base adducts, potentially disrupting the DNA base paring. However, adduct formation at the phosphor-ribose backbone was proposed long ago and has recently been shown for DNA-phosphate adducts formed by the tobacco-specific nitrosamine 4-(methylnitrosamino)-1-(3-pyridyl)-1-butanone [96].

### 3.2. DNA Adductomics

The emergence of nontargeted adductomics will generate a huge amount of DNA adduct data, presumably representing the exposome. Studies of true DNA adductomics are limited, but first reports are very promising. For example, applying a nontargeted ‘omics’ approach, thousands of potential DNA adduct *m*/*z* features were observed in human tissues such as lung, bronchia, or saliva [97,98,99]. A challenging task will be to accurately identify the low abundant DNA adducts resulting from exogenous sources that are dwarfed by epigenetic marks and endogenous DNA adducts that are present at levels several orders of magnitude higher [100,101]. 

### 3.3. DNA Adduct Mapping

The second question in the DNA adduct field was, and still is, to determine the location of exposure-induced DNA adducts. Therefore, the mapping of B[*a*]P-derived DNA adducts has been reported genome-wide [60] and for specific genes such as *P53* [102], *kRAS*, and *hRAS* [103], suggesting DNA adduct formation at mutation hotspots in these genes. B[*a*]P treatment however has been shown to also induce oxidative stress, which is known to cause multiple DNA lesions recognized by the UvrABC Nuclease, potentially confounding the results [104]. Elegant stable isotope-labeled experiments confirmed the preferred binding of BPDE to the mutation hotspot sequence motif in *TP53* [105]. 

Further, several approaches have been developed for genome-wide mapping of 8-oxo-dG with various degrees of resolution (reviewed by Poetsch [58]). The mapping revealed accumulation of 8-oxo-dG at sites of high nucleosome occupancy in yeast, and different types of GC repeats accumulate large amounts of 8-oxo-dG, particularly telomeres and microsatellites [54], suggesting that DNA adduct formation is not random across the genome. These DNA adduct mapping studies highlight the importance of the DNA adduct location to elucidate subsequent biological outcomes. 

### 3.4. Single Molecule DNA Sequencing

The current nanopore-type Single Molecule DNA Sequencing technologies do not reach the chemical specificity obtained by mass spectrometry, but efforts are underway to combine nanopore with mass spectrometry [106,107,108]. Further, since the ion signal disturbances are caused by physio-chemical properties of the DNA molecule transitioning the nanopore, refined artificial intelligence (AI)-assisted data analyses may allow chemically specific identifications of the DNA adducts at any location in the genome. This novel technology is still in the implementation phase, and future studies are needed to evaluate and establish chemical specificity, sensitivity, and accuracy for measuring DNA adducts. It is conceivable that the ONT and PacBio platforms are unsuitable for measuring interstrand crosslinks because they are both sequencing single-stranded DNA. Therefore, efforts and developments are on the way to use wider pores to perform sequencing of double-stranded DNA [109,110,111]. Crosslinks of DNA to whole proteins will probably not work with this new technology.

## 4. Conclusions

With the current technologies on hand, it is easy to determine whether DNA adducts form in the target tissue to establish the internal dose derived from external exposure. When the goal is to understand the exposome, mass spectrometry-based adductomics is the method of choice. Detection and quantitation of DNA adducts derived from a mixture of pollutants in the target tissue will unambiguously demonstrate, with high chemical specificity, that the subject has been exposed and that the toxicant has reached the tissue of concern. 

If the goal is to understand how exposure leads to changes in cell homeostasis and promotes or prevents disease development, many DNA adduct mapping approaches are available. These mapping methodologies will enable the investigator to demonstrate, with biological specificity, that the exposure or treatment induces modifications in the DNA at the promotor or gene region of interest. 

The new *Single Molecule DNA Sequencing* technologies are expected to be suitable for addressing both questions mentioned above simultaneously. They will provide a comprehensive picture of the DNA adducts types, levels, and their locations across the nuclear and mitochondrial genome. 

Lastly, the previous paradigm of increased DNA adducts being equal to increased risk for mutations and cancer need to be revisited since DNA adduct locations may be the driving indicator for cancer risk. If so, we need a better understanding of site-specific effects of DNA adducts on DNA structure and function, epigenetic marks, and subsequent gene expressions associated with disease development. Therefore, a novel approach is needed for meaningful interpretation of the DNA adductome-based exposomes and future DNA adduct maps to improve our understanding of cancer etiology and explain the origin of mutational signatures established for various tumor types [112]. 

## Data Availability

Not applicable.

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
