# Peer review of "Current and Future Methodology for Quantitation and Site-Specific Mapping the Location of DNA Adducts"

_toxics, 2022, doi:10.3390/toxics10020045_

Round 1
Reviewer 1 Report
Toxics
“Current and future methodology for…”
Boysen and Nookaew
This is a very interesting, and timely communication giving an overview of the status of a number of cutting edge approaches for assessing DNA damage. The manuscript is well written, and generally comprehensive (for its length). However, the manuscript would benefit from a number of revisions, and further consideration of a number of points:
Title (and throughout). ‘quantification’, and quantify would be better used, as was used in the Abstract (and make the use consistent).
Abstract
The formation of… …cancer risk.
Reiner and Zamenhof (no initials)
L35 …the binding to DNA has… (no comma)
L35 …epidemiology studies, in a variety of biological matrices
L40 …all DNA adducts. Not quite all, some DNA adducts remain undetectable by DNA adductomic approach. For example, only recently were DNA-DNA crosslinks capable of being detected by a DNA adductomic approach (Hu et al. (2019) DNA crosslinkomics: a tool for the comprehensive assessment of inter-strand crosslinks using high resolution mass spectrometry. Analytical Chemistry. 91, 15193-), and it remains unclear whether pyrimidine dimers would be detected by existing adductomic methodology.
L44 …highlights
Methodology
L48. a reference to cover the immunochemical detection of DNA adducts is needed e.g., Rossner et al. (2016) Urinary 8-oxo-7,8-dihydro-2'-deoxyguanosine analysis by an improved ELISA: does assay standardization reduce inter-laboratory variability? Free Radic. Biol. Med. 95, 169-179.
L64. Here and elsewhere, 2’-deoxyribonucleosides and nucleobases are more precise terms
L67. It is highly recommended/necessary that stable isotope internal standards are added to all targeted assays (this is impossible for the untargeted assays, for every analyte, although use of some standards is recommended to account for matrix effects etc., see: Chang et al. (2018) Novel approach to integrated DNA adductomics for the assessment of in vitro and in vivo environmental exposures. Archives of Toxicology. 92, 2665-).
L72. ‘exposome’
L75 onwards. The results from untargeted assays are unlikely to be able to be accurately expressed as units per mg DNA or number of unmodified dN (not least due to absence of internal standards).
…nucleotides should be 2’-deoxyribonucleosides
2.3
CPD have also been detected using damaged DNA immunoprecipitation (see ref 21).
L95. Define 8-oxo-dG (preferably 8-oxodG), and 5-hm-dC (5-hmdC?), not least as 8-oxodG is used again, where it is defined.
Limitations of these assays also include the limited availability of antibodies with suitable specificity for adducts of interest
L105. Why is ‘Single’ and ‘Molecule’ capitalized, but not ‘sequencing’?
L112. nucleobases
L135 nucleobases
L139 and 141. 5-mdC
L141. Define these adduct abbreviations
L142. Already defined
Figure 1 legend – please provide more detail/description of what is being demonstrated, and how the data were derived.
L177. Define ‘nnt’
L203. …benzo
L208. One adduct per 10 human cells – is this value correct? …in lung cells…
L212. 8-oxodG should have been defined earlier.
L213. “Oxidatively generated modifications of DNA…” (this is the preferred terminology, for the rationale, see: Cadet et al. (2012) Biologically relevant oxidants and terminology, classification and nomenclature of oxidatively generated damage to nucleobases and 2-deoxyribose in nucleic acids. Free Radic. Res. 46, 367-).
This sentence is incorrect. Oxidatively generated modifications of DNA (specifically 8-oxodG, no other modifications have been examined to any meaningful extent) were never thought to be entirely artefacts generated during sample work-up and analysis. However, it was acknowledged that the discrepancies in levels of damage determined by different methods was due to artefact, and this could lead to an over-estimation of the true levels [see: ESCODD: Collins, A; Gedik, C; Vaughan, N; Wood, S; White, A; Dubois, J; Duez, P; Dehon, G; Rees, J.F; Loft, S; Moller, P; Poulsen, H; Riis, B; Weimann, A; Cadet, J; Douki, T; Ravanat, J.L; Sauvaigo, S; Faure, H; Morel, I; Morin, B; Epe, B; Phoa, N; Hartwig,A; Pelzer, A; Dolara, P; Casalini, C; Giovannelli, L., Lodovici, M; Olinski, R; Bialkowski, K; Foksinski, M; Gackowski, D; Duraková, Z; Hliniková, L; Korytar, P; Sivonová, M; Duinská, M; Mislanová, C; Viña, J; Lloret, A; Möller, L; Hofer, T; Nygren, J; Gremaud, E; Herbert, K; Chauhan; Kelly, F; Dunster, C; Lunec, J; Cooke, M; Evans, M; Patel, P; Podmore, I; White, A; Wild, C; Hardie, L; Olliver, J; Smith, E. (2002) Comparative analysis of baseline 8-oxo-7,8-dihydroguanine in mammalian cell DNA, by different methods in different laboratories: an approach to consensus. Carcinogenesis, 23, 2129-].
L214. …in part still true... this is not really the case as those aware of the field take the necessary precautions to avoid artefact. Nonetheless, the issue of artefactual formation of damage does still arise as, over time, the lessons from the work of ESCODD (see ref 82, and above), have been forgotten.
L215. The point the authors are trying to make is reasonable, although comparing levels of one adduct, in one cell type, to another adduct in a different cell type, does not make for the basis of a strong argument. Furthermore, 8-oxodG is expected (and has to some extent been reported) to be widely present in many cell types – this may not be the case for N2-BPDE-dG.
On the basis that the authors are suggesting that the formation of these adducts is a rare, are they suggesting that most cancers are derived from endogenous exposures (which seems to run counter to the thinking of Rappaport et al.)?
Furthermore, this is an argument that supports the measurement of these adducts in urine, where the body burden of exposure can be determined, and DNA repair and urinary excretions serves to ‘concentrate’ adduct levels, compared to those seen in DNA. [DNA adductomics has been successfully applied to the analysis of adducts in urine, see: Cooke et al. (2018) Urinary DNA Adductomics – A Novel Approach for Exposomics. Environment International. 121, 1033-]
L228. …lung, bronchia, saliva and urine… (add above ref)
L228 onwards. Low abundance adducts from exogenous sources vs. epigenetic (see spelling) marks and endogenous adducts. True, but is it not possible that exogenous exposures affect the epigenetic marks and endogenous adduct levels, as part of the carcinogenic process? If so, then this adds yet further weight to examining the totality of adducts via adductomics.
L240. …settling this discussion… can the authors please elaborate, what discussion, and what is settled? The point of this first paragraph is not entirely clear.
L247. Importance of DNA adduct location – this is a valuable point although, to a large extent, nuclear DNA has been the sole focus with only few studies examining the distribution of adducts in mitochondrial DNA (see ref 21, for example).
Conclusions
L258. it is, not it’s
L261. …is the method of choice. ‘the’ or ‘a’ – are there suitable alternatives to DNA adductomics to achieve this goal?
L264. Can adduct levels also inform on (disease e.g., cancer) risk – the more adducts, the greater the risk?
L269. Mapping techniques can only imply downstream effects from location e.g., hotspots of damage correspond to hotspots of mutation, but what about the other consequences of adducts (discussed in ref 53)?
L271. …picture of the DNA adduct types, levels and locations across the nuclear and mitochondrial genomes.
Author Response
"Please see the attachment."

Reviewer 2 Report
This Communication summarized recent technologies utilized in DNA adduct analysis, emphasizing the single-molecule DNA sequencing for mapping the DNA adduct locations. However, I feel this Communication needs to strengthen the description of the applications of single-molecule DNA sequencing. There are very interesting researches mentioned very briefly in the Discussion. For example, as an emerging and complementary technique to mass spectrometry (MS), the combined efforts of nanopore with MS, in my opinion, should be elaborated in more detail. This may be the future for DNA adduct analysis and deserves recognition.
The descriptions of MS-based targeted and untargeted approaches for analyzing DNA adducts in Sections 2.1 and 2.2 were grossly insufficient and inaccurate. For example, these two sections did not distinguish which approach is quantitative or qualitative and why. I understand the space is limited for a Communication, but at least the author can point to specific reviews for the audience to get more detailed information.
Line 57, the first two sentences under Section 2.1 are misleading. This sentence implied that electrospray ionization that revolutionized biochemistry was introduced by Dole in 1968. But it was Fenn in the 1990s that applied Electrospray ionization (ESI) of intact protein in MS that revolutionized biochemistry. In addition, “electrospray mass-spectrometry” is not a proper term. Electrospray ionization is a type of ionization source that is widely used nowadays, yet there is a limit to the type of molecules this ion source can ionize. Certainly, ESI can not analyze “any type of compound,” as was claimed in line 59. Please rewrite these sentences.
Here are some other issues that need to be addressed.
Section 1, please remove “Chemical carcinogenesis” at the beginning of the first paragraph. This section did not discuss anything else. Thus the sub-section title is not needed.
Page 1 line 43, first paragraph, this is a Communication, thus the sentence should start with “this communication” instead of “this perspective”
The first paragraph under Section 2. Methodology needs improvement. It did not mention any technologies utilized in Scheme 1. And Scheme 1 needs better descriptions in the text.
Line 73, MS/MS is tandem MS, not data dependent MS.
Line 129, it should be “technically,” not “technical”
Page 4 around line 139. These are examples of DNA adducts detected by ONT, thus should be moved to section 2.3.1.1
I suggest the authors make a table to summarize the detection of DNA adducts by the nanopore technology, specifying if the source of DNA was in vitro or in vivo, or from humans or not.
Line 31, to keep consistency, the initial for Zamenhof’s first name should be given.
Line 35, remove the comma after “the binding to DNA.”
Line 237, it should be “is” not “his” in this sentence
Author Response
"Please see the attachment."

Reviewer 3 Report
This is a very good perspectives on adductomics and state-of-the art on localization of adduct by nanopore sequencing. The latter is a cutting edge and ambitious, albeit preliminary.
- Line 211-218; 8OHdG has been detected in much larger amount the BPDE adducts, but detection sensitivity and organ specificity should be argued. Relatively bulkier adduct like PhIP, reports of detection in human tissue have been still few and reproducibility especially in terms of quantitation is still a challenge. Add the authors' comments on the organs (tissues; PBC vs visceral organs, sensitivity of the methodologies, LC-MSMS vs immunological section vs postlableing)
- Provide the areas of uncertainties and challenges such as locations of larger m/w adduct in the genome when applying Single molecule DNA sequencing and the list of previously available species of DNA adduct by this method.
Author Response
"Please see the attachment."

Round 2
Reviewer 2 Report
Thanks for addressing the reviewers' comments.
Please correct the sentence starting from line 74. There is no "electrospray mass-spectrometry", and the electrospray ionization cannot analyze "any type of compounds."
Please double-check the resolution of all figures. They all look blurry in version 2.
Author Response
The ESI statemen has been revised to read now
"... Electrospray ionization coupled with mass-spectrometry is now a commonly used technique for qualitative and quantitative analyses of many type of compounds,..."
We also improved the resolution of Scheme 1